

# Antimicrobial peptides properties beyond growth inhibition and bacterial killing

Israel Castillo-Juárez[1], Blanca Esther Blancas-Luciano[2], Rodolfo García-Contreras[2] and Ana María Fernández-Presas[2]

[1] Laboratorio de Fitoquímica, Posgrado de Botánica, Colegio de Postgraduados, Texcoco, Estado de México, Mexico
[2] Departamento de Microbiología y Parasitología, Facultad de Medicina, Universidad Nacional Autónoma de México, Mexico City, Mexico City, Mexico

## ABSTRACT

Antimicrobial peptides (AMPs) are versatile molecules with broad antimicrobial activity produced by representatives of the three domains of life. Also, there are derivatives of AMPs and artificial short peptides that can inhibit microbial growth. Beyond killing microbes, AMPs at grow sub-inhibitory concentrations also exhibit anti-virulence activity against critical pathogenic bacteria, including ESKAPE pathogens. Anti-virulence therapies are an alternative to antibiotics since they do not directly affect viability and growth, and they are considered less likely to generate resistance. Bacterial biofilms significantly increase antibiotic resistance and are linked to establishing chronic infections. Various AMPs can kill biofilm cells and eradicate infections in animal models. However, some can inhibit biofilm formation and promote dispersal at sub-growth inhibitory concentrations. These examples are discussed here, along with those of peptides that inhibit the expression of traits controlled by quorum sensing, such as the production of exoproteases, phenazines, surfactants, toxins, among others. In addition, specific targets that are determinants of virulence include secretion systems (type II, III, and VI) responsible for releasing effector proteins toxic to eukaryotic cells. This review summarizes the current knowledge on the anti-virulence properties of AMPs and the future directions of their research.

# INTRODUCTION

The discovery of antibiotics is one of the most important events in modern medicine. The scientific community interest and the pharmaceutical industry for their commercialization in the mid-20th century favored the so-called golden age of these molecules (*Díaz-Nuñez, García-Contreras & Castillo-Juárez, 2021*). However, the generation of resistance of microorganisms to bactericides is a global public health problem and represents one of the critical challenges to be solved by humanity (*Muñoz Cazares et al., 2017*). Therefore, new targets or mechanisms of action are being investigated, in which antimicrobial peptides (AMPs) are an option to combat drug-resistant infections (*Boparai & Sharma, 2019*; *Lei et al., 2019*; *Magana et al., 2020*).

Corresponding authors
Rodolfo García-Contreras,
rgarc@bq.unam.mx
Ana María Fernández-Presas,
presas@unam.mx

Most living organisms produce antimicrobial peptides as a defense mechanism in eukaryotes or as a microenvironmental competition strategy in prokaryotes (*Moretta et al., 2021*). Around 17,363 AMPs have been described, in which 82.7% are synthetic, and the rest are produced naturally in the three domains of life (*Bulet, Stöcklin & Menin, 2004*; *Boparai & Sharma, 2019*; *Zasloff, 2019*). They are classified according to their source of origin, activity, structure, and amino acid composition (*Huan et al., 2020*). Most AMPs are monomers of 4 to 50 amino acids that can acquire an amphipathic secondary structure of $\alpha$-helix, $\beta$-hairpin-like $\beta$-sheet, $\beta$-sheet, or $\alpha$-helix/ $\beta$-sheet mixed structures (*Bulet, Stöcklin & Menin, 2004*).

In mammals, AMPs are a fundamental part of the innate immune system to counteract microbial infections (*Boman, 2000*). Some, such as defensins, are produced by epithelial cells to prevent the establishment of pathogens and are generally found in phagocytic cells to help eliminate microorganisms when ingested (*Bulet, Stöcklin & Menin, 2004*; *de la Fuente-Núñez et al., 2017*). In plants, AMPs are produced in different tissues to protect against pathogens; specifically, thionins and snakins are the best known (*Tang et al., 2018*). Bacteriocins are AMPs produced by bacteria, which have been identified as having a high antimicrobial activity (*Soltani et al., 2021*), while in archaea, halocins and sulfolobicins are the two main classes of archaeocins, which meet several ecological functions of competition in the environment with extreme conditions (*Besse et al., 2015*).

Classical antimicrobial properties are the main characteristic described for AMPs, and they are active against a broad spectrum of microorganisms, including viruses and parasites (*Harris, Dennison & Phoenix, 2009*; *Huan et al., 2020*). The primary mechanism reported for AMPs is related to their ability to lyse microbial cells (*Pasupuleti, Schmidtchen & Malmsten, 2012*; *Mankoci et al., 2019*) since the cationic properties (net positive charge) of most of them allows them to interact with the membranes of microorganisms (*Alghalayini et al., 2019*) (Fig. 1). However, other action mechanisms have also been described in which AMPs interact directly with specific target molecules (*Brogden, 2005*; *Le, Fang & Sekaran, 2017*; *Graf & Wilson, 2019*). Some of them have similar action mechanisms to antibiotics, including the inhibition of protein synthesis (pleurocidin and indolein) (*Subbalakshmi & Sitaram, 1998*; *Patrzykat et al., 2002*), or cell wall synthesis (mersacidin) (*Brötz et al., 1998*). Others, such as temporin L and the synthetic peptide 35409 (RYRRKKKMKKALQYIKLLKE), inhibit *Escherichia coli* divisome machinery (*Barreto-Santamaría et al., 2016*; *Di Somma et al., 2020*). Unfortunately, because AMPs affect the viability of microorganisms, resistance mechanisms towards them are also reported (*Cassone et al., 2009*; *Haney, Straus & Hancock, 2019*).

In addition, AMPs influence several other biological processes (*Haney, Straus & Hancock, 2019*); for example, they interfere with the regulation of the microbiota, wound healing, induction of adaptive immunity, as well as possess anti-inflammatory, pro-inflammatory, anti-cancer, and cytotoxic properties, among others (*Beisswenger & Bals, 2005*; *Haney, Straus & Hancock, 2019*; *Huan et al., 2020*). Thus, due to its multifunctional nature, some authors have begun to use the broader term ''host defense peptide'' (HDP) (*Haney, Straus & Hancock, 2019*).

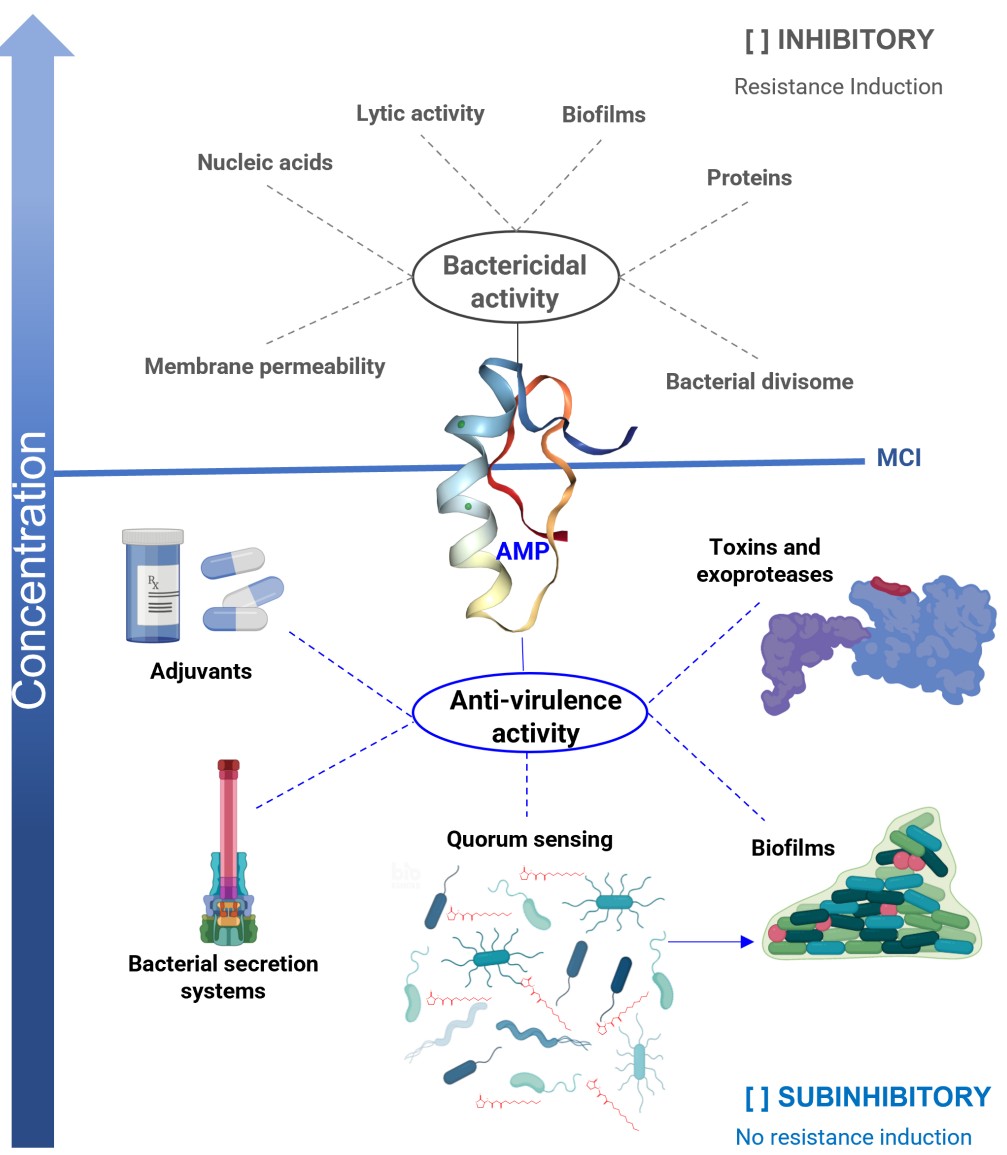

**Figure 1  Antibacterial properties of antimicrobial peptides (AMP).** The bactericidal properties are one of the main characteristics of AMP, its lytic capacity being one of the best-studied mechanisms of action. However, other targets have been identified in which they act as nucleic acids, proteins, or the divisome machinery. Unfortunately, as with other bactericidal agents, they also induce resistance. When AMPs work at sub-inhibitory concentrations, they exhibit anti-virulence properties, reducing the production of various factors that cause damage, but without affecting the viability of the bacteria. One of the targets is the inhibition of quorum sensing (QS), a general regulator of virulence. Furthermore, AMPs inhibit bacterial secretion systems, inactivate toxins, and exhibit adjuvant properties, restoring the activity of antibiotics on resistant strains. In the anti-biofilm activity, AMPs can act by bactericidal mechanisms or antivirulence by inhibiting QS. An ideal property for anti-virulence therapies is that they do not generate resistance or are expected to do so to a lesser degree. MIC = minimum inhibitory concentration.

Anti-virulence activity is the antimicrobial property that has been discovered in various molecules when used at growth sub-inhibitory concentrations, in which they block the ability of bacteria to cause damage without interfering with their viability (*Castillo-Juarez et al., 2017*). There are different anti-virulence targets, but the most studied are the quorum sensing (QS) systems (*Jiang et al., 2019*) and the type 3 secretion systems (T3SS) (*Hotinger & May, 2019*).

QS is a phenomenon of gene regulation at the population level dependent on bacterial density that allows bacteria to exhibit collective or multicellular behaviors (*Díaz-Nuñez, García-Contreras & Castillo-Juárez, 2021*). It is one of the best-studied anti-virulence targets because it regulates the expression of various virulence factors, including the formation of biofilms, which is a multicellular behavior that gives them high resistance to antimicrobials (*FleitasMartínez et al., 2018*; *Jiang et al., 2019*).

In this regard, it is reported that some AMPs also exhibit anti-virulence properties at sub-inhibitory concentrations. In which the inhibition of biofilms (*Di Somma et al., 2020*), QS systems (*Overhage et al., 2008*), and secretion systems (*McShan & De Guzman, 2015*) stand out. They also neutralize enzymes, such as exoproteases and toxins (*Kudryashova, Seveau & Kudryashov, 2017*; *Gusman, Malonneet & Atassi, 2001*). In addition, they are reported to have adjuvant properties, which help restore the bactericidal effect of antibiotics on resistant strains (Fig. 1) (*Geitani et al., 2019*).

This review focuses on describing and analyzing the anti-virulence properties of AMPs exhibited in sub-inhibitory concentrations described so far, highlighting the evidence of their possible application.

## Survey methodology

To ensure an inclusive and unbiased analysis of literature and to accomplish the review's objectives, a comprehensive analysis of published articles on the activity of antimicrobial peptides using the following online databases: Medline (PubMed), Science Direct (http://sciencedirect.com) database, Web of Science, Scopus, and Google Scholar system. Additionally, the following keywords were used: antimicrobial peptides, anti-virulence properties, quorum sensing, biofilms, targets together with Boolean operators such as "AND" and "OR".

## Anti-biofilm and anti-quorum sensing activity of AMPs

Biofilms are the preferred lifestyle of bacteria and are structured microbial aggregates, surrounded by a self-produced extracellular matrix, and attached to biotic or abiotic surfaces. Biofilms are involved in most chronic bacterial infections (*Bjarnsholt, 2013*). Moreover, they are crucial determinants of bacterial virulence. The biofilm matrix is formed by diverse components present in the extracellular polymeric substances: mainly proteins, polysaccharides, extracellular nucleic acids, and ions (*Donlan, 2002*). Biofilm formation is an ordered process, beginning with the initial contact and attachment to surfaces, mainly mediated by structures such as flagellum and fimbria, followed by micro-colony formation, maturation, and formation of the complex biofilm architecture, finally, detachment and dispersal of some cells from the biofilm occur (*Sutherland, 2001*).

Biofilms are pivotal for bacterial survival as they protect against adverse environmental conditions. They increase drug resistance by various mechanisms such as the decrease in the permeability of antibiotics, the promotion of dormancy and induction of bacterial persistence, the expression of the efflux pumps of antibiotics, and the synthesis of periplasmic glucans (aminoglycosides) that inactivate antibiotics (*Hall & Mah, 2017*). Biofilms also allow bacteria to evade the human defense mechanisms (*Mirzaei et al., 2020*) since several biofilm matrix proteins protect biofilms against human innate immune cells, opsonization, and phagocytosis (*Lewis, 2008*). Moreover, it has been demonstrated that some bacterial species, previously known as extracellular pathogens, can reside inside various host cells by adapting to intracellular life through the formation of microbial aggregates similar to bacterial biofilms, leading to their long-term survival inside the cells (*Mirzaei et al., 2020*).

Unlike antibiotics, AMPs are suitable for slowing growth and killing cells in the biofilm. Several examples of effective AMPs with this activity have been described that correlate with the ability of AMPs to resolve bacterial infections *in vivo*. For a recent full review of these activities and the translation potential of such peptides, see the work of Gislaine and coworkers (*Silveira et al., 2021*).

Since the aim of this work is to discuss the anti-virulence potential of AMPs, and one of the premises of anti-virulence therapies is not to affect directly bacterial growth and survival, most of the examples of AMPs with anti-biofilm activity discussed here will be peptides that inhibit biofilm formation at growth sub-inhibitory concentrations (Table 1).

AMP activity against biofilms is mediated by the degradation or destabilization of the extracellular matrix (*Yasir, Willcox & Dutta, 2018*). The PI peptide (derived from polyphemusin I) induces the degradation of the exopolysaccharides produced by *Streptococcus mutans*, causing the biofilm formation to be attenuated (*Zhang et al., 2019*). Also, an AMP complex produced by the insect *Calliphora vicina* promotes the degradation of the matrix of the biofilm produced by *E. coli*, *Staphylococcus aureus*, and *Acinetobacter baumannii* (*Gordya et al., 2017*). Hepcidin 20 from the human liver decreases the extracellular matrix and disrupts the architecture of *S. epidermidis* biofilms (*Brancatisano et al., 2014*). S4 (1-16) M4Ka (dermaseptin S4 derivative), which inhibits immature biofilms of *Pseudomonas fluorescens* (*Quilès et al., 2016*). Piscidin-3 is derived from fish, which degrades the extracellular DNA of *P. aeruginosa* biofilms (*Libardo et al., 2017*).

Biofilm inhibition by AMPs is also mediated by the downregulation of genes responsible for biofilm formation and transport of binding proteins; for example, in Staphylococcal biofilms, the $\beta$-defensin 3 from humans decreases the expression of the *icaA*, *icaD*, and *icaR* genes that codify enzymes responsible for the biosynthesis of the adhesin PIA, essential for biofilm formation (*Rohde et al., 2010*; *Zhu et al., 2013*). In addition, AMPs also inhibit genes that control the transport and binding proteins, such as ABC transporters that are involved in biofilm formation since they promote cell-to-surface and cell-to-cell interactions (*Zhu et al., 2013*; *Wang et al., 2017*).

**Table 1  Main peptides and polypeptides with anti-virulence and adjuvant properties.**

| Name | Source | Activity | Effect | References |
|---|---|---|---|---|
| PI peptide (Derived from polyphemusin I) | Horseshoe crab | Anti-biofilm | Inhibits the development of biofilm of *S. mutans* in the dental plaque of rabbit incisors. | *Zhang et al. (2019)* |
| Hepcidin 20 | Derived from human liver | Anti-biofilm | Inhibit the production and accumulation of extracellularmatrix in the biofilm of *S. epidermidis*. | *Brancatisano et al. (2014)* |
| AMP complex (defensin, cecropin, diptericin and proline rich peptide families) | *Calliphora vicina* worms | Anti-biofilm | Destroys the matrix and cells of the biofilmv of *E. coli*, *S. aureus*, and *A. baumannii*. | *Gordya et al., 2017* |
| S4 (1-16) M4Ka (Dermaseptin S4 derivative) | Amphibian skin | Anti-biofilm | Destroys immature *P. fluorescens* biofilms. | *Quilès et al., 2016* |
| Piscidin-3/(Cu$^{2+}$) | Fish | Anti-biofilm | Damages *E. coli* DNA in a copper-dependent manner. | *Libardo et al., 2017* |
| $\beta$-defensin 3 | Humans | | Decreases the formation of biofilms in *Staphylococcus*, as well as the expression of genes responsible for its production. | *Zhu et al. (2013)* |
| LL-37 (Derived from cathelicidin) | Humans | Anti-biofilm, anti-QS | Reduces the expression of the *Las* and *RhI* genes. Inhibits the biofilm formation in *P. aeruginosa*, *F. novicida*, *S. epidermidis*, and *S. aureus*. | *Hancock & Sahl (2006)*; *Overhage et al. (2008)*; *Chennupati et al. (2009)*; *Amer, Bishop & van Hoek, 2010*; *Hell et al., 2010*; *Kang, Dietz & Li, 2019*. |
| LIVRHK and LIVRRK | Synthetics | Anti-QS, anti-biofilm | They inhibit biofilm formation and the production of virulence factors (pyocyanin, protease, and rhamnolipids) in *P. aeruginosa*. Also, they reduce the expression of *lasI*, *lasR*, *rhlI*, and *rhlR*. | *Taha et al., 2019* |
| Peptide 1037 | Synthetic | Anti-biofilm | Inhibits the formation of biofilms of *P. aeruginosa*, *B. cenocepacia*, and *L. monocytogenes*. Also, it reduces the expression of a variety of genes involved in its formation. | *De La Fuente-Núñez et al. (2012)* |
| D-Bac8c$^{2,5Leu}$ | Synthetic | Anti-biofilm | Prevents the formation of *S. aureus* biofilms on catheters. | *Zapotoczna et al. (2017)* |
| Bovicin HC5 | *Streptococcus bovis* HC5 | Anti-biofilm, anti-QS | Reduces the formation of biofilms in *S. aureus*. | Pimentel-Filho Nde et al., 2014 |
| Nisin | *Lactococcus lactis* | | | |
| Subtilosin | *Bacillus subtilis* KATMIRA1933 | Anti-biofilm, anti-QS | Reduces the production of violacein in *C. violaceum*. Also, biofilm formation and AI-2 production in *G. vaginalis*. | *Algburi et al., 2017* |

**Table 1** (*continued*)

| Name | Source | Activity | Effect | References |
|------|--------|----------|--------|------------|
| RBP15 | Synthetic | Anti-QS | Inhibits the phosphorylation of the RNAIII activator protein (TRAP) in *S. aureus*. | *Yang et al. (2003)* |
| P1(EWESDNRLNEEQ) and P2 (TKLTRTWRQ) | Synthetic | Anti-T2SS | They disrupt the XcpVW pseudopilin nucleus complex and the tip of the pseudopilus. Inhibit T2SS and reduce the virulence of *P. aeruginosa* in the *Caenorhabditis elegans* model. | *Zhang et al. (2018)* |
| Lactoferrin | Mammals | Anti-T3SS | Inhibit T3SS in *Salmonella*, *Shigella*, and *E. coli* through the degradation of translocon proteins. | *McShan & De Guzman (2015)* |
| CoilA, Coil B and CesA2 | Synthetic | Anti-T3SS | Inhibit the formation of the T3SS needle in EPEC and reduce hemolysis. | *Larzábal et al. (2019)* |
| HNP, HD5 | Human | Anti-toxin | Inhibit the Lethal Factor of *B. anthracis*, diphtheria toxin, exotoxin A of *P. aeruginosa* and cytotoxin B of *C. difficile*. | *Kim et al. 2005*, *2006*; *Giesemann, Guttenberg & Aktories (2008)*; |
| hBD | Human | Anti-toxin | Inhibits the gonococcal toxin NarE of *N. gonorrhoeae* and the Lethal Factor of *B. anthracis*. | *Rodas et al., 2016*; *Wei et al., 2009* |
| Retrocyclins | Human | Anti-toxin | Inhibit the Lethal factor of *B. anthracis* and the vaginolysin of *G. vaginalis*. | *Wang et al. (2006)*; *Hooven et al. (2012)* |
| Bacitracin | *Bacillus subtilis* | Anti-toxin | They inhibit various toxins such as Lethal Factor (*B. anthracis*), C2 toxin (*C. botulinum*), CDT transferase (*C. difficile*), and epsilon toxin (*C. perfringens*). | *Schnell et al. (2019)* |
| Histatin 5 | Human | Anti-toxin | Inhibits the exoproteases of *P. gingivalis* involved in the generation of damage in periodontal disease and the cysteine proteinases of *C. histolyticum*. | *Gusman et al., 2001*; *Le, Fang & Sekaran (2017)* |
| Unarmycin A and C | Marine bacteria | Adjuvants | Inhibit the azole antifungal efflux pumps and restore antifungal sensitivity in *C. albicans*. | *Tanabe et al. (2007)* |
| Plantaricin PLNC8 $\alpha\beta$ | *Lactobacillus plantarum* | Adjuvants | Enhances the activity of conventional antibiotics against *Staphylococcus* strains. | *Bengtsson et al. (2020)* |

**Notes.**

**T3SS**, type 3 secretion system; **T2SS**, type 2 secretion system; **QS**, quorum sensing; **EPEC**, enteropathogenic *E. coli*.

Beyond inhibiting biofilm maturation, AMPs can also inhibit initial attachment and increase cell dispersal. One of the first discovered AMPs with the ability to eradicate biofilms was LL-37 (*Overhage et al., 2008*), derived from human cathelicidin, an amphipathic peptide widely distributed in body fluids (*Burton & Steel, 2009*). At low concentrations, LL-37 inhibits the adhesion of *P. aeruginosa* cells to surfaces, and at higher concentrations, it reduces the thickness of the biofilms (*Hancock & Sahl, 2006*). Moreover, LL-37 also eradicates *P. aeruginosa* biofilms *in vivo* (*Chennupati et al., 2009*). The anti-biofilm effects of LL-37 in *P. aeruginosa* at concentrations that do not affect viability and growth are related to the upregulation of the expression of type IV pili genes that lead to the promotion of twitching motility which is linked to biofilm dispersal and to the decrease in the expression of flagellar genes which leads to lower attachment to surfaces (*Overhage et al., 2008*). In addition, LL-37 treatment induces a strong down-regulation of the core genes of the *Las* and *Rhl* QS systems and the repression of genes that encode QS-dependent virulence factors such as LasB elastase and those responsible for the biosynthesis of rhamnolipids (*Overhage et al., 2008*). In addition, it inhibits the biofilm formation of other pathogens such as *Francisella novicida* and *S. epidermidis* (*Amer, Bishop & van Hoek, 2010*; *Hell et al., 2010*).

Other AMPs can prevent biofilm formation by inhibiting quorum sensing (*Overhage et al., 2008*). For example, Trp-containing peptides inhibit QS-regulated virulence and biofilm growth of multidrug-resistant P. aeruginosa. Significantly, peptides containing tryptophan at low concentrations reduced the production of virulence factors that regulate the gene expression of the Las and Rh1 systems. Biofilm formation was inhibited in a concentration-dependent manner, which was associated with inhibiting extracellular polysaccharide production by negatively regulating the transcription of *pelA*, *algD*, and *pslA*. These changes were correlated with alterations in the extracellular production of virulence and motility.

Also, two novel synthetic peptides (LIVRHK and LIVRRK) can inhibit biofilm formation of *P. aeruginosa* PA01, and QS-dependent phenotypes such as pyocyanin exoprotease, and rhamnolipid production were identified. In addition, a down-regulation of the expression of the core QS genes *lasRI* and *rhlRI* were observed, corroborating the inhibition of QS (*Taha et al., 2019*).

The discovery of the anti-biofilm and anti-QS properties of LL-37 led to the search for other natural and synthetic peptides with similar properties. De la Fuente and his colleagues in 2012 selected 50 small synthetic peptides and identified 16 with anti-biofilm activity against *P. aeruginosa*, with HH15 being one of the best. According to their sequence, 15 small peptides were designed, including peptide 1037 of only nine amino acids, reducing the biofilm formation of *Burkholderia cenocepacia* and *Listeria monocytogenes* (*De La Fuente-Núñez et al., 2012*).

The peptide 1037, like LL-37, stimulates twitching motility and decreases the expression of flagellar genes, leading to potent inhibition of swimming and swarming motilities (*De La Fuente-Núñez et al., 2012*). Comparison between the effect in gene expression of peptides (LL-37 *vs.* 1037) allowed the identification of ten common downregulated genes and four upregulated ones. The role of those genes in biofilm formation was confirmed using

transposon mutants of each one, being nine of the ten mutants in the downregulated genes lower biofilm producers than the parental strain. Although the involved genes mainly were hypothetical proteins, the flagellar gene *flgB*, *rhlB*, involved in rhamnolipid biosynthesis and *nirS* encoding a nitrite reductase were identified. In addition, two of the four mutants in the upregulated genes (a hypothetical protein and *actP*, encoding an acetate permease have) higher biofilm producer than the parental strain (*De La Fuente-Núñez et al., 2012*).

In another study, it was shown that LL-37 exhibits anti-biofilm activity against *S. epidermidis*, where at low concentrations they prevent cell attachment, while at high concentrations, they prevent the maturation and establishment of biofilms. Also, LL-37 has a potent *S. aureus* biofilm eradication activity (*Kang, Dietz & Li, 2019*).

Other synthetic peptides such as D-Bac8c$^{2,5Leu,}$ D-HB43, and D-ranalexin have effectively killed *S. aureus* biofilms. For example, the synthetic peptide D-Bac8c$^{2,5Leu}$, when applied as a catheter lock solution, has inhibitory activity on early and mature *S. aureus* biofilms in a rat venous catheter infection model (*Zapotoczna et al., 2017*).

Although some classic antibiotics at sub-MIC concentrations have shown anti-virulence and QS system regulation behaviors (*Skindersoe et al., 2008*; *Zhang & Li, 2016*), there is still little research on peptide antibiotics. However, bovicin HC5 (broad-spectrum lantibiotic) and nisin (polycyclic peptide antibiotic) have been reported to have anti-biofilm activity through QS interference from *S. aureus* (Pimentel-Filho et al., 2014). Similarly, subtilosin (cyclic lantibiotic) reduces violacein production in *Chromobacterium violaceum* (indicative of QS inhibition), as well as biofilm formation and autoinducer-2 (AI-2) production in *Gardnerella vaginalis* (*Algburi et al. al., 2017*) (Table 1).

## Other anti-virulence targets of AMPs

Although biofilms and other QS-controlled phenotypes (exoproteases, phenazines, rhamnolipids, swarm motility) are essential for bacterial virulence, some important virulence factors are not positively regulated by QS. Eight secretion systems have been found in Gram-negative and Gram-positive bacteria. However, these systems are sometimes unregulated by QS or may even be downregulated, as in some *vibrio* species. Therefore, in cases where QS negatively regulates them, QS inhibition can promote virulence through secretion systems (*Pena et al., 2019*). Therefore, specific inhibitors of these systems in combination with QS inhibitors may be necessary to develop more robust antibacterial therapies (*García-Contreras, 2016*).

Accordingly, Zhang and coworkers elucidated the structural and functional details of the pseudopilus tip complex of the type II secretion system of *P. aeruginosa*, which is an essential component of the system that functions as a piston, allowing the export of multiple effectors (*Zhang et al., 2018*). Based on the structural details of the complex, two mimicking peptides [P1(EWESDNRLNEEQ) and P2 (TKLTRTWRQ)] that were able to compete with the binding of the XcpV and XcpW pseudolipins were designed, retaining the specific amino acids that allow the interaction between those pseudopilins and introducing other hydrophilic amino acids to enhance solubility. The utilization of those peptides precluded the formation of the core complex essential for the pseudopilus tip formation and strongly attenuated secretion through the type II secretion system (*Zhang et al., 2018*).

Interestingly, natural mammalian peptides, such as iron-binding lactoferrin, are potent inhibitors of T3SS in enteric bacteria (*Salmonella, Shigella,* and *E. coli*) by inducing translocon protein degradation. This activity is mediated by its binding to the lipopolysaccharide on the bacterial surface, destabilizing the protein-protein interactions essential for the system. Furthermore, lactoferrins have serine protease activity that can affect T3SS cleavage proteins (*McShan & De Guzman, 2015*).

Beyond the anti-T3SS of natural peptides, the strategy of using polypeptides that mimic some components of the systems and that compete with the binding of the natural bacterial components was effective to inhibit the system in Chlamydia, Salmonella, and Shigella, blocking their entrance to eukaryotic cells in cultures (*McShan & De Guzman, 2015*). Similarly, in enteropathogenic *E. coli* (EPEC), coiled-coil peptide mimetics, analogs of the EspA, EscF, and CesA proteins (CoilA, Coil B and CesA2) of its T3SS, inhibit the T3SS mediated hemolysis (*Larzábal et al., 2019*).

Another critical virulence determinant is the type VI secretion system, which delivers multiple effectors to prokaryotic and eukaryotic cells. Those effectors target cell walls, cell membranes, DNA, and to avoid self-poisoning, bacteria that produce them also produce neutralizing proteins that bid the effectors. Recently, in *P. aeruginosa*, the effector TplE, a lipolytic toxin effective against other bacteria and able to disrupt the endoplasmic reticulum in eukaryotic cells, had been characterized; this protein is neutralized by TplEi (*Jiang et al., 2016*).

Based on this interaction, Gao and coworkers generated a small peptide capable of competing with the TplEi-TplE interaction, for this TplE was hydrolyzed, generating a 26 amino acid fragment that strongly binds with TplEi, releasing the TplE toxin, and thus inducing the autointoxication of *P. aeruginosa* (*Gao et al., 2017*). This approach is attractive and represents a new concept for generating new inhibitors of secretory systems and other potential targets. A similar approach was recently used for the identification of small peptides that inhibit antitoxins that belong to the toxin-antitoxin systems (*Lee et al., 2015*; *Sundar, Rajan & Piramanayagam, 2019*), which are related to latency, persistence (*Page & Peti, 2016*) and bacterial virulence (*Fernández-García et al., 2016*). These systems are abundant in intracellular bacterial pathogens such as *Mycobacterium tuberculosis* (*Sala, Bordes & Genevaux, 2014*).

Additional anti-virulence activities of some AMPs, such as defensins are the capacity to bind and inhibit the activity of several bacterial toxins and related virulence factors (Table 1). Defensins are components of the innate immunity of mammals and are also found in invertebrates, plants, and fungi. Although these peptides had low sequence similarity, they share common structural features and display broad antibacterial and antiviral activity at high concentrations; in addition, they modulate inflammation and promote angiogenesis and wound healing. Moreover, they can neutralize several bacterial toxins, among them cytolysin, listeryolysin that promote pore formation, ribosyltransferase toxins, glycosylation promoting toxins, the MARTX toxins from Vibrio and Aeromonas, the Panton-Valentine leucocidin, staphylokinase from *S. aureus*, SIC which is the Streptococcal inhibitor of complement (*Kudryashova, Seveau & Kudryashov, 2017*). Upon binding to the toxins, defensins promote their unfolding, disrupting their secondary and tertiary structure,
making them more susceptible to proteolysis and promoting their precipitation. Although the physicochemical properties that allow defensins to bind and neutralize a broad range of structurally diverse toxins are not completely understood, recent studies demonstrate that defensins act by recognizing regions of proteins showing structural plasticity and thermodynamic instability, features that are shared by a wide range of bacterial toxins (*Kudryashova, Seveau & Kudryashov, 2017*). In general, of the alpha-class such as HNP and HD5, they inhibit the lethal factor of *Bacillus anthracis*, the diphtheria toxin, the exotoxin A of *P. aeruginosa*, the cytotoxin B of *Clostridioides difficile*, among others (*Kim et al. 2005*, *2006*; *Giesemann, Guttenberg & Aktories, 2008*). While those in the beta-class, such as hBD, inhibit the gonococcal toxin NarE from *Neisseria gonorrhoeae* and the lethal factor from *B. anthracis* (*Rodas et al., 2016*;*Wei et al., 2009*). In the case of those of the theta-class, the retrocyclins inhibit the lethal factor of *B. anthracis* and the vaginolysin of *G. vaginalis* (*Wang et al., 2006*; *Hooven et al., 2012*).

Beyond defensins, there are other notable examples of toxin-neutralizing peptides, such as the artificial peptide Pep19−2.5 and related ones, capable of inactivating lipopolysaccharides (LPS or endotoxin) and lipoproteins *in vitro* and *in vivo*. They also decrease inflammation mediated by the activation of signaling cascades (*Heinbockel et al., 2018*), and their efficacy has been reported in several mouse infection models, including endotoxemia and bacteremia (*Heinbockel et al., 2013*). Several other peptides with the ability to neutralize a wide variety of bacterial toxins have been described, for which we recommend consulting the following reviews (*Jerala & Porro, 2005*; *Kudryashova, Seveau & Kudryashov, 2017*; *Schnell et al., 2019*). The effect was not always determined at sub-MIC concentrations; however, the inhibition of toxins is a strategy contemplated within the anti-virulence targets.

Bacitracin is an antibiotic that inhibits cell wall synthesis, but recently it has also been reported to neutralize type A/B protein exotoxins by inhibiting pore formation, preventing translocation of the A subunit to the host cell cytosol. These toxins are made up of an enzymatic component (A subunit) and a binding/transport component (B subunit), such as the lethal factor of *Bacillus anthracis*, the toxin C2 of *Clostridium botulinum*, the CDT transferase of *C. difficile*, and epsilon toxin of *Clostridium perfringens* (*Schnell et al., 2019*). In addition to neutralizing bacterial toxins, some AMPs can inhibit exoproteases implicated in the generation of host damage during periodontal disease. For example, the salivary peptide histatin 5 inhibits the host metalloproteases and exoproteases from bacterial pathogens such as the gingipains produced by *Porphyromonas gingivalis* attenuating damage and inflammation (*Gusman, Malonneet & Atassi, 2001*). Moreover, histatin 5 also inhibits cysteine proteinases such as clostripain, which is produced by *Clostridium histolyticum* during gangrene, while other AMPs inhibit exoproteases such as subtilisin A, proteinase K, elastase, and chymotrypsin (*Le, Fang & Sekaran, 2017*).

Finally, a characteristic of some anti-virulence molecules is their adjuvant properties, which enable them to restore the activity of antibiotics on resistant strains (*Díaz-Nuñez, García-Contreras & Castillo-Juárez, 2021*). This strategy is very promising, and although it does not prevent the generation of resistance, it allows the reactivation of antimicrobials that are in danger of falling into disuse (*González-Bello, 2017*). In the case of AMPs at

sub-MIC concentrations, some reports of adjuvant properties have been made in the literature, such as unarmycin A and C (*Tanabe et al., 2007*). These cyclopeptides isolated from marine bacteria are azole antifungal ejection pump inhibitors and restore fluconazole sensitivity of resistant strains and clinical isolates of *Candida albicans* (*Tanabe et al., 2007*). Also, plantaricin PLNC8 $\alpha$ $\beta$ showed an adjuvant effect by potentiating the activity of conventional antibiotics (vancomycin, rifampicin, and gentamicin) against *S. epidermidis*, although the mechanism of action involved is unknown (*Bengtsson et al., 2020*).

## CONCLUSIONS

There is currently enough evidence to support the participation of the QS, T3S, two-component regulatory systems, and other virulence determinants in the generation of bacterial pathogenicity and damage (*Marshall & Brett Finlay, 2014*; *Totsika, 2016*; *Tiwari et al., 2017*; *Tsai et al., 2020*). It is reported that the interruption of genes that code for these systems reduces virulence and bacterial pathogenicity *in vivo* models of animals and plants (*Castillo-Juárez et al., 2015*; *Jiang et al., 2019*). Also, similar results are obtained with the administration of small molecules that inhibit these systems (*Marshall & Brett Finlay, 2014*; *Jiang et al., 2016*; *Hotinger & May, 2019*). Similarly, there are reports of the anti-virulence properties of synthetic peptides analogous to the autoinducers of Gram-positive bacteria, such as the so-called RIP and its derivatives (RBP15), which reduce pathogenicity at the preclinical level (*Yang et al., 2003*; *Singh, Desouky & Nakayama, 2016*). However, to achieve the implementation of anti-virulence therapies in the clinical practice, there are some challenges to overcome, such as determining their toxicity, their possible side effects, including their effects on the microbiota, the generation of resistance, and verifying their efficacy at the clinical level (*Díaz-Nuñez, García-Contreras & Castillo-Juárez, 2021*).

The anti-virulence effects of substances at low concentrations has generated significant interest due to the possibility of controlling microbial infections and probably avoiding the appearance of resistance (*Totsika, 2016*; *Díaz-Nuñez, García-Contreras & Castillo-Juárez, 2021*). Various substances, including natural products, antibiotics, and drugs of mass consumption such as ibuprofen and aspirin have been identified to reduce virulence at sub-MIC concentrations (*Bernardo et al., 2004*; *Skindersoe et al., 2008*; *El-Mowafy et al., 2014*; *Soo et al., 2017*; *Dai et al., 2019*). The information related to the effect of low-dose AMPs is scarce and highly debatable. Recently, evidence that indicates adverse effects of the use of AMP at sub-inhibitory doses was compiled. The possible adverse effects include the induction of resistance (strong stress generators) and directly or indirectly stimulating virulence through different signaling pathways (*Vasilchenko & Rogozhin, 2019*). It should be noted that the main characteristic of the ideal anti-virulence molecule is that it does not interfere directly with bacterial viability, lowering the selection pressure for the generation of resistance (*Díaz-Nuñez, García-Contreras & Castillo-Juárez, 2021*). Most peptides stress bacterial cells at growth inhibitory concentrations; however, there is evidence to suggest that in their native environment, peptides are in relatively low concentrations that do not kill microorganisms, and hence the high growth inhibitory concentrations are hardly reached (*Dorschner et al., 2001*; *Monnet, Juillard & Gardan, 2016*; *Vasilchenko & Rogozhin,*

*2019*). Therefore, the ubiquitous microbicidal activity of AMPs may not be their primary natural or ecological function.

Also, it has been pointed out that AMPs exhibit anti-virulence properties, but the effect is unpredictable and possibly uncontrollable since AMPs also may stimulate virulence at specific concentrations (*Vasilchenko & Rogozhin, 2019*). In this regard, hormesis is a widely studied phenomenon that occurs at low doses, in which the same compound can exhibit antagonistic effects depending on the dose (*Mattson, 2008*). This phenomenon has been described in some small and synthetic molecules, but it is not a generality for all the anti-virulence molecules described. In the case of AMPs, a possible case of hormesis of the LL-37 peptide is pointed out, which at sub-MIC concentrations reduces the gene expression of QS and the production of virulence factors, but at the same time stimulates others (*Overhage et al., 2008*; *Strempe et al., 2013*). In hormesis, concentration is essential to obtain the desired effect; however, identifying peptides that can stimulate QS systems could be helpful if applied to bacteria that regulate the expression of beneficial phenotypes. As in the case of beneficial microorganisms in agriculture, for the treatment of wastewater or the intestinal microbiota (*Schikora, Schenk & Hartmann, 2016*; *Zhang & Li, 2016*; *Bivar Xavier, 2018*). In this sense, nanotechnological techniques will be essential to help to maintain their bioavailability efficiently (*Boparai & Sharma, 2019*).

With the information available to date, some behaviors, or effects of peptides at sub-MIC concentrations can be classified as autoinducer peptides, inducer peptides, signal peptides, and anti-virulence peptides (Fig. 2). Autoinducer peptides are produced by Gram-positive bacteria [AIP (autoinducing peptide), CSP (competence stimulating peptide), ComX, and CSF (competence and sporulation factor)] and participate in bacterial communication through QS systems (*Monnet, Juillard & Gardan, 2016*). In comparison, the inducer peptides are those that are produced by other microorganisms (antibiotics, bacteriocins) or the host (defensins) and that modify the gene expression of the QS systems or virulence (*Baishya et al., 2021*). The signaling peptides are produced by host cells for specific functions (such as promoting the establishment of beneficial microorganisms of the intestinal microbiota), but bacteria also capture them as environmental signals for regulating virulence systems. Finally, anti-virulence peptides are produced by the hosts (or by competing microorganisms) as a strategy to reduce pathogenicity and avoid the establishment and damage of bacteria (Fig. 2). It should be noted that one and two-component environmental signaling systems can participate in all these effects (*Tiwari et al., 2017*).

Some authors mention the term "pheromone" to refer to the induction of gene expression "at a distance" (distances are challenging to define in the microscopic world) by specific peptides (*Monnet, Juillard & Gardan, 2016*; *Yajima, 2016*; *Vasilchenko & Rogozhin, 2019*). However, we consider it a confusing term and suggest that it should be avoided in this study topic as it is based on an analogy of the functioning of pheromones in macroscopic organisms with sexual reproduction. Likewise, in the classification by activity of anti-virulence peptides, their possible effects on host cells should be considered (*Tornesello et al., 2018*) as well as their immunogenic activity, which some authors point out is the

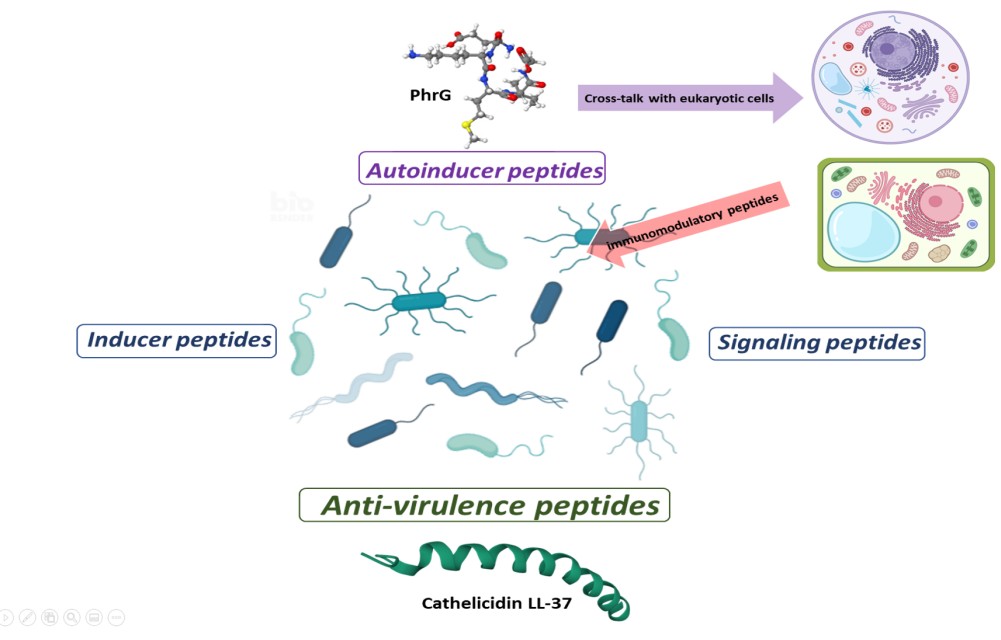

**Figure 2** **Proposal for peptide role at sub-MIC concentrations in microbial populations.** Gram-positive bacteria produce autoinducer peptides for bacterial communication by quorum sensing. These can interfere with eukaryotic cells and induce adverse or beneficial effects. In the same way, cells can generate peptides as an immunogenic response to combat pathogenic microorganisms (AMPs or anti-virulence). Anti-virulence peptides can also be produced as a competition strategy within microbial populations. A particular microbial population does not produce inducer peptides, but they manage to alter their gene expression. The effect of these peptides seems to be random and a consequence of the peptides that circulate within the complex communication network. In turn, signal peptides allow a bacterial population to perceive when microenvironmental conditions are adequate or inappropriate and generate a reaction. These can favor the establishment or dispersal of populations.

main responsible for eliminating bacteria *in vivo* (*Hancock & Sahl, 2013*; *Mansour, Pena & Hancock, 2014*).

On the other hand, although the U.S. Food and Drug Administration (FDA) has approved several AMPs used at growth inhibitory concentrations, most are restricted to the topical application due to limitations found with other routes of administration, such as a short half-life, low stability, and low bioavailability (*Lei et al., 2019*). Likewise, its use at inhibitory concentrations for prolonged periods is reported to generate toxic effects, like hemolysis, and to induce resistance (*Rathinakumar, Walkenhorst & Wimley, 2009*; *Starr et al., 2018*; *Lei et al., 2019*), coupled with the high cost of producing them commercially (*Moretta et al., 2021*). Investigating the activities of AMPs at low or growth sub-inhibitory concentrations could help resolve some of these difficulties and favor their clinical application. Therefore, we can conclude that the action of AMPs and the response they elicit at sub-MIC concentrations is a fertile and promising area of knowledge that requires further research to develop safe and effective anti-virulence therapies.

Finally, regardless of the anti-biofilm and anti-virulence properties of the peptides discussed here, some aspects should be further tested, including their utilization to

attenuate infections produced by clinical strains *in vivo*, and more significant efforts for their implementation in clinical trials should be encouraged (*Silveira et al., 2021*).

Another aspect that needs to be studied further is the mechanistic details of the action of peptides that inhibit biofilm formation (without affecting bacterial growth), virulence factors controlled by QS, and secretion systems. It is essential to study the possibility of selecting resistance *in vivo*, its effects, and the mechanisms involved. Although it is proposed that the peptides are more robust and less likely to induce resistance compared to the usual antibiotics, some possible mechanisms are proposed, such as modifications of the membrane and the composition of the cell wall, expulsion by efflux pumps, AMP sequestration, and protease inactivation (Assoni et al., 2020). In this regard, it is expected that similar mechanisms will eventually evolve to attenuate the effects of AMP on biofilm and inhibit virulence, even if these peptides do not affect viability and growth *in vitro* in principle.

It was especially considering that although scarce, resistance mechanisms against anti-virulence therapies, mediated by QS inhibition (*García-Contreras, 2016*) and biofilm inhibition, had been described (*Travier et al., 2013*).

### Funding

Ana María Fernández-Presas is funded by PAPITT, DGAPA, UNAM, Mexico City, grant #IN218419, Rodolfo García-Contreras is funded by CONACYT grant CB 2017-2018 number A1-S-8530 and by PAPITT UNAM grant number IN214218. Israel Castillo Júarez is funded by Cátedras-CONACyT program. Blanca Esther Blancas-Luciano is supported by CONACYT grant # 424031 for her doctoral studies. The funders had no role in study design, data collection and analysis, decision to publish, or preparation of the manuscript.

### Grant Disclosures

The following grant information was disclosed by the authors:
PAPITT, DGAPA, UNAM, Mexico City: #IN218419.
CONACYT grant: CB 2017-2018 number A1-S-8530.
PAPITT UNAM: IN214218.
Cátedras-CONACyT program.
CONACYT: # 424031.

### Competing Interests

The authors declare there are no competing interests.

### Author Contributions

- Israel Castillo-Juárez analyzed the data, prepared figures and/or tables, authored or reviewed drafts of the paper, and approved the final draft.
- Blanca Esther Blancas-Luciano, Rodolfo García-Contreras and Ana María Fernández-Presas analyzed the data, authored or reviewed drafts of the paper, and approved the final draft.

## Data Availability

This is a literature review.

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
