# Peer review of "Antimicrobial peptides properties beyond growth inhibition and bacterial killing"

_PeerJ, doi:10.7717/peerj.12667_

## Round 0.1 · original submission · Minor Revisions

Dear authors:

Thank you so much for the excellent submission to Peer J. The two reviewers agree that the manuscript is suitable for publications, only minor revisions should be done in order to accept the manuscript. I am returning the manuscript so you can amend these minor aspects of the paper and once returned it will not go to further review.

Congratulations and thank you for submitting this excellent manuscript to PeerJ.

Warm regards

Reviewer 1 ·

Basic reporting

The article is very well written and clear, It deserves the addition of a table, it would complement the article.
I recommend that the table contain the peptides that inhibit
biofilm formation at sub-inhibitory concentrations of growth, peptide sequence, mechanisms of action, tests against which microorganisms and test development phase

Experimental design

no comment

Validity of the findings

no comment

Additional comments

no comment

Reviewer 2 ·

Basic reporting

This review is interesting. the authors show recent information about these antimicrobial peptides. the subject is of interest to the microbiological community. i have some minor comments.

in the section "Other anti-virulence targets of AMPs" I suggest adding headings on the topics type III secretion systems, bacterial toxins and other targets to facilitate the reader's better understanding of AMPs.

In the conclusions section, figure 2 is cited before figure 1, which is not orthodox.

Other minor comments include :


53 type III, III means IV?

99 typo: delete ")"

100 typo: delete ")"

195 please define : EPS

197 for the reader clarity, replace "calliphora vicina", for from the insect calliphora vicina.


199 clarify "S. epidermis" is Streptococcus or Sthaphylococcus ?

200 please clarify for first time is P. aeruginosa, Pseudomonas aeruginosa?


239 replace PA01 for Pseudomonas aureginosa PA01

245 the reference: "De la fuente and colleagues 2012", does not exist in the references

249 typo: burkhodetia

254-261 please cite these results

271 define CLS

315-316 please cite these results

366 is B. anthracis? Bacillus


392 define TC system


470 define FDA

Experimental design

no comment

Validity of the findings

no comment

Additional comments

no comment

---

## Round 0.2 · accepted · Accept

Dear authors:

I am pleased to inform you that after careful revision of the submitted files, the paper ‘Antimicrobial peptides properties beyond growth inhibition and bacterial killing’ has been accepted for publication. I thank authors for considering the comments done by the two experts that review your paper and providing a solid review, that will be of interest to a broad audience.

Best regards